# Feasibility and Preliminary Efficacy of Enhanced Midwifery Care to Support Women Experiencing Subclinical Depression: A Pilot Randomised Controlled Trial

**DOI:** 10.3390/ijerph22121835

**Published:** 2025-12-08

**Authors:** James R. John, Wendy Pickup, Antonio Mendoza Diaz, Sara Cibralic, Aleisha Heys, Virginia Schmied, Bryanne Barnett, Valsamma Eapen

**Affiliations:** 1School of Clinical Medicine, University of New South Wales, Level 3 AGSM, Sydney, NSW 2052, Australia; james.john@unsw.edu.au (J.R.J.);; 2Academic Unit of Infant, Child, and Adolescent Psychiatric Services, South Western Sydney Local Health District, Liverpool, NSW 2170, Australia; 3Ingham Institute of Applied Medical Research, Liverpool, NSW 2170, Australia; 4Tasmanian Centre for Mental Health Service Innovation, Hobart, TAS 7000, Australia; 5School of Nursing and Midwifery, Western Sydney University, Campbelltown, NSW 2560, Australia; 6St. John of God Healthcare, Sydney, NSW 2134, Australia

**Keywords:** subclinical depression, enhanced midwifery care, continuity of care, parenting confidence, CALD

## Abstract

**Highlights:**

**Public health relevance—How does this work relate to a public health issue?**
Subclinical depression in the perinatal period is common and under-recognized, and may increase risk for later clinical depression and adverse outcomes for mothers and infants.This study focuses on women from ethnically diverse communities in South Western Sydney, where barriers to mental health and maternity care are well documented.

**Public health significance—Why is this work of significance to public health?**
This pilot RCT shows that enhanced midwifery care can be delivered with good fidelity in a hard-to-reach population, underscoring its potential to reduce perinatal mental health inequities.The findings provide early, implementation-relevant evidence to inform scalable, integrated models of perinatal mental health support within routine antenatal care.

**Public health implications—What are the key implications/messages for practitioners, policy makers, and researchers?**
The findings of this study reiterate the importance of early identification and proactive support for women with subclinical depressive symptoms, highlighting the need to strengthen early intervention pathways in antenatal care.Integrating enhanced midwifery continuity with multidisciplinary support may represent a promising, feasible framework to strengthen engagement among vulnerable groups; adequately powered trials are needed to determine effectiveness and optimise retention.

**Abstract:**

This study investigated the feasibility and preliminary effects of enhanced midwifery care in reducing subclinical depression symptoms among women in ethnically diverse areas of the South Western Sydney Local Health District (SWSLHD). A pilot randomised controlled trial was conducted among pregnant women attending the Fairfield and/or Liverpool antenatal clinic with an Edinburgh Depression Scale (EDS) score of 10–12 (i.e., just below the generally accepted clinical cut-off score of 13 to indicate subclinical depressive symptoms) during the first antenatal visit (i.e., before 26 weeks gestation). Participants were randomly allocated to either the intervention group which received continuous and coordinated support from a dedicated Registered Midwife (RM) trained in counselling and linked with a multidisciplinary team, or the usual care group, which received standard maternity care from various providers without continuity or additional coordinated support. Primary outcomes included feasibility of recruitment, randomisation, intervention delivery and fidelity, and retention and follow-up. The secondary outcomes were improvement in depressive symptom severity assessed via EDS, psychological distress (Kessler’s psychological distress scale—K10), and parenting confidence (Karitane Parenting Confidence Scale (KPCS). Descriptive analyses were used to assess the feasibility outcomes, whereas mixed-effects models were used to examine the effects of treatment on secondary outcomes. Thirty-seven mothers were recruited into the study, of which eighteen were randomised to the intervention group and nineteen to the usual care group. The intervention was delivered with good fidelity, and remote adaptations during COVID-19 ensured both continuity of care and high retention at 6-month follow-up. Findings of the mixed-effects models showed significant within-group reduction in EDS scores over time, with scores at 8 weeks postpartum (T2) significantly lower than at baseline (T0; β = −2.77, SE = 1.36, *p* < 0.05) but no significant differences between the groups (β = −0.02, SE = 1.63, *p* = 0.992) or time-by-group interactions at any timepoint for EDS, K10, and KPCS. These findings demonstrate the feasibility of enhanced midwifery care in a “hard to reach” population of SWSLHD and highlight the need for adequately powered trials to determine its effectiveness on maternal mental health and parenting outcomes.

## 1. Background

Antenatal depression during pregnancy is a prevalent concern among expectant mothers and has significant implications for maternal and foetal health [1,2]. The prevalence of antenatal depression varies widely across studies but has been reported to affect between 10 and 20% of pregnant women [3]. These rates are twice as high among women from culturally and linguistically diverse (CALD) backgrounds compared to women from non-CALD backgrounds due to a combination of pre-migration (e.g., persecution) and post-migration (e.g., isolation, language barriers, trust with health care system) factors [4,5]. Evidence shows that undetected anxiety and depression are common and can adversely affect a mother’s physical health and psychosocial wellbeing while also increasing the risk of psychological, developmental, and neurocognitive difficulties in the child [6,7]. Besides health ramifications, the cost of perinatal depression in Australia amounts to AUD 877 million per year, including for health system use, productivity loss, and poor wellbeing, with additional lifetime costs up to AUD 5.2 billion [8]. Therefore, early identification and tailored intervention supports are essential in reducing both the health and economic burdens associated with antenatal depression.

The antenatal period offers a unique opportunity for early intervention and support to prevent and/or limit any further exacerbation of negative outcomes. However, current reviews have highlighted gaps in evidence on maternal health interventions, particularly noting the underutilisation of midwives and the lack of culturally tailored approaches for migrant, refugee, and culturally diverse women, emphasising the need for strategic approaches to reduce disparities in perinatal outcomes [9,10]. In view of this, several governmental initiatives have been implemented, one of which is the New South Wales (NSW), Australia, SAFE START Strategic Policy [11] aimed at providing universal psychosocial assessment, depression and anxiety screening, and follow-up care and support during the perinatal period (period of time from pregnancy to 1 year postpartum). In this program, routine screening of pregnant women for current or potential mental health problems, specifically anxiety and depression, is mandated throughout maternity services and risk is assessed using the Edinburgh Depression Scale (EDS) [12]. It is recommended that the EDS is administered twice during pregnancy and once postnatally, with appropriate care being provided to women with high depressive and anxiety symptoms (i.e., EDS scores of 13 or more) [13].

Universal psychosocial screening during the initial antenatal assessment has been shown to be critical for identifying risk early and enabling the provision of relevant supports [14]. However, these models are often constrained by inadequate staff training, unaffordable antenatal classes, and limited referral options, resulting in case reviews and referrals being restricted to women with the highest depression and anxiety scores (EDS ≥ 13). Women who experience anxiety, have low to moderate depression, have low psychological resources, and/or indicate that they need support but do not have high EDS scores often fall between the service gaps [15,16]. This is concerning as women who score in the subclinical range on the EDS (scores 10–12), that is, within approximately one standard deviation below the commonly used clinical cut-off, may not meet criteria for clinical depression but still experience some level of depression, requiring attention before complexities set in [17]. Hereafter, we refer to this group as experiencing subclinical depression symptoms. This concern is exacerbated for women from CALD backgrounds, who are less likely to be screened due to language barriers [18], under-report their mental health concerns when screening is provided, and seek help when challenges are and identified [19]. Hence, identifying and supporting women with subclinical depression symptoms and providing care to these women during pregnancy is important, as these women remain at heightened risk of developing postnatal depression and anxiety if left unsupported.

Emerging evidence has shown that the enhanced midwifery care model is effective in improving maternal mental health during the perinatal period [20]. This model involves continuity of care by a midwife or group of midwives, care coordination, targeted education and care, and ongoing monitoring and support of health and wellbeing during pregnancy and the postnatal period [21,22,23]. For example, a review of 15 trials including 17,674 women found that midwife-led continuity models of care were associated with reduced interventions at birth, improved maternal satisfaction, and showed a trend toward cost savings compared to other care models [22]. Additionally, a recent systematic review [20] of six articles found that enhanced midwifery care led to decreased anxiety and depression symptoms in mothers during the perinatal period. Further, an international review [24] of midwifery continuity of care initiatives across multiple countries found that, in Australia, 23 programs have been implemented for various vulnerable groups including young parents, Indigenous women, and those experiencing rural, remote, or socio-economic disadvantage but have not specifically targeted women from CALD backgrounds.

To address the knowledge gap, this study aimed to examine the feasibility and preliminary effects of enhanced midwifery care on improving depressive symptoms, psychological distress, and parenting confidence among pregnant women from CALD communities in the South Western Sydney Local Health District (SWSLHD) who presented with subclinical depressive symptoms. Findings of this study will help inform larger scale evaluations to inform health care policy and practice to better meet the mental health needs of pregnant women, thereby promoting healthier pregnancies and improved maternal, neonatal, and family outcomes, particularly among priority population groups.

## 2. Methods

### 2.1. Study Design, Sites, and Participants

Study findings are reported in accordance with the consolidated standards of reporting trials (CONSORT) guidelines [25]. A multisite, single-blinded, pilot randomised controlled trial (RCT) was employed to compare the effects of enhanced midwifery care versus usual care. This trial was registered with the Australian New Zealand Clinical Trials Registry (ACTRN12623000698673) on 30 June 2023.

Inclusion criteria were pregnant woman under 26 weeks gestation with a competent level of English, attending the Fairfield and/or Liverpool midwifery clinics in the SWSLHD with subclinical depressive symptoms (EDS score of 10–12) during the antenatal visit. Further, women who were over 26 weeks gestation, who planned to give birth at a different hospital, or those without competent levels of English were excluded from this study.

Potentially eligible participants were identified via antenatal booking records and approached via phone, email, or face to face while at the antenatal clinic by the Research Midwife (RM) and asked whether they would like to hear more about the study. During COVID-19 lockdown, clinic schedules were screened daily; those who were interested received the study information and informed consent was collected electronically through REDCap.

### 2.2. Randomisation and Allocation Concealment

A 1:1 ratio of randomisation was followed, whereby each eligible participant was randomised to either the intervention or control group using the REDCap Randomisation Module. The REDCap Randomisation Module is a process that assigns participants by chance (rather than by choice) into specific groups and is concealed from the recruiters [26]. Participants were not stratified by risk profile. Allocation in REDCap was assigned after all the participants’ baseline information was obtained alongside signed electronic consent. Following randomisation, the RM and participants were not blinded to the type of care because the nature of the intervention made concealment impractical, whereas the Biostatistician remained blinded during outcomes analysis to minimise bias.

### 2.3. Intervention and Control Groups

Women in enhanced midwifery care (intervention group) received continuous, coordinated support from a dedicated, specially trained RM who liaised with a multidisciplinary team, offered personalised care planning, and provided ongoing psychosocial support from pregnancy through the postnatal period. In contrast, women in usual care (control group) received standard maternity care delivered by different midwives and clinicians across perinatal and postnatal stages, without dedicated continuity or additional coordinated support.

Enhanced midwifery care: Enhanced midwifery care was led by a dedicated RM with over 30 years of experience, providing comprehensive care coordination and navigation from pregnancy through the early postnatal period as well as clinical and counselling experience. The RM worked closely with a multidisciplinary team including the obstetric team, social workers, Child and Family Health Nurses (CFHNs), General Practitioners (GPs), Allied Health, and mental health staff, to ensure coordination of care. All women in the intervention group had direct access to the RM and could reach out at their antenatal visits, text message, email, or a phone call, allowing for flexible and personalised support. Services were drawn from a pool of non-pharmacological interventions shown to have some efficacy in preventing postpartum anxiety/depression or increasing service provision and engagement. This included access to individualised care planning, home visiting, telephone support, the offer of debriefing after birth, social support, and flexible care provision, including outreach [27,28,29,30,31,32].

Usual care group: Women in the usual care group received usual care - that is standard maternity services provided by a range of midwives, obstetricians, and other medical or allied health professionals throughout pregnancy, birth, and the postnatal period. In contrast to the intervention group, care was delivered by different clinicians at each stage, with no dedicated continuity from a single provider, and without additional coordinated support or proactive follow-up to address emotional, social, or mental health needs.

During the study, women in either group who received a high score for any of the assessments performed at each timepoint were contacted and escalated via routine clinical pathway and adequately supported.

### 2.4. Data Collection and Outcomes

This study involved the delivery of questionnaires at four timepoints (T0: recruitment-before 26 weeks’ gestation; T1: 36 weeks’ gestation; T2: 8 weeks post-partum; and T3: 6-month follow-up) to evaluate women’s experience of care across the two groups.

### 2.5. Primary Outcome

Primary outcomes included measures of feasibility, assessed through recruitment (number of participants enrolled versus approached), randomisation (ability to allocate participants and maintain group integrity), intervention fidelity (completion of planned contacts with the RM), and retention (proportion of participants completing follow-up assessments collected across all timepoints).

### 2.6. Secondary Outcomes

The Edinburgh Depression Scale (EDS) is a short assessment that was administered at the antenatal visit with the RM who had access to antenatal booking appointments. The assessment comprise 10 short statements, yielding a total score ranging from 0 to 30, with scores of 13 and higher being indicative of likelihood of meeting criteria for a psychiatric diagnosis of depression [12]. Scores between 10 and 12 are considered to indicate subclinical depression risk, with an indication to “watch and wait”. The EDS was collected at four timepoints (T0, T1, T2, and T3). The EDS has also been used to assess level of anxiety [33] with three very specific anxiety-related questions [34].

Kessler’s Psychological Distress Scale (K10) is a 10-item standardised self-reported measure of anxiety and depressive symptoms over the past four weeks [35]. Total scores range from 10 to 50, with scores of 10–19 regarded as likely to be well; 20–24 as likely to have a mild disorder; 25–29 as likely to have a moderate disorder; and 30–50 as likely to have a severe disorder. The K10 was administered at T1, T2, and T3.

Karitane Parenting Confidence Scale (KPCS) was used to determine parenting self-efficacy in parents of infants during the first year [36]. The assessment comprises 15 items with scores ranging from 0 to 45, where a score of 39 or less indicates lower confidence levels. This assessment was collected at T2 and T3.

The sociodemographic information collected included mother’s age (in years), country of birth (Australia/Other), ethnicity (Caucasian, non-Caucasian), language spoken at home (English/Other), CALD status (composite of non-Caucasian and/or language other than English), indigenous status (Yes/No), smoking status (Yes/No), and number of weeks pregnant at the time of enrolment.

### 2.7. Sample Size Calculation

Using an α = 0.05 and a β = 0.08, with two groups measured over three timepoints (assuming a correlation between repeated measures of 0.5 and applying a non-sphericity correction), a repeated-measures analysis of variance (ANOVA) indicated that a total sample of 34 participants would provide sufficient sensitivity to detect a moderate effect (f = 0.25). This level of sensitivity was judged to be adequate for a pilot study, which is not intended to maximise its specificity but rather serves as a proof of concept demonstrating one group will show different changes over time than another.

### 2.8. Data Analysis

Data management was facilitated through the REDCap online platform. Descriptive analyses such as an independent-samples *t*-test (for continuous variables) and chi-square tests or Fisher’s exact tests (for categorical variables) were used to determine any significant differences in demographic and clinical characteristics across the treatment groups. Feasibility indicators were summarised as follows: recruitment was assessed as the proportion of participants enrolled versus approached; randomisation was evaluated by the ability to allocate participants and maintain group integrity; intervention fidelity was measured by adherence to the enhanced midwifery care model, specifically the completion of planned contacts with the RM, using structured fidelity checklists to document each contact; and retention was calculated as the proportion of participants completing follow-up assessments, with differences between groups further assessed using Fisher’s exact test. Further, intention-to-treat (ITT) linear mixed-effects models with Bonferroni correction for multiple corrections were used to examine the effects of treatment on secondary outcomes. Linear mixed-effects models can effectively account for within-subject correlations and handle unbalanced data, making them well-suited for analysing RCT data with missing values [37]. Fixed effects such as time, treatment group, and time by group interaction were also assessed separately for each outcome variable whilst adjusting for sociodemographic covariates. Significant effects were followed up with pairwise contrasts comparing group differences in changes from baseline to follow-up. *p*-values less than 0.05 were considered statistically significant. All analyses were undertaken in Statistical Package for the Social Sciences (SPSS) version 28 (IBM SPSS, IBM Corp., Armonk, NY, USA) and R studio version 4.1.2.

## 3. Results

### 3.1. Participant Characteristics

Baseline demographic and clinical characteristics of the participants by treatment group are shown in Table 1. The sample was ethnically diverse, with two in three (65%) identifying as from a CALD background. The average age of the participants was 30 years, and they were on average 19 weeks pregnant at the time of enrolment. The baseline characteristics of the sample were similar, with no significant differences between the intervention and control groups.

### 3.2. Feasibility Outcomes

Recruitment: The recruitment and follow-up period was between 10 May 2021 and 30 April 2023. Of the 305 participants assessed for eligibility, 37 mothers were eligible and consented to participate. The study successfully engaged women with subclinical depressive symptoms from CALD background, demonstrating the ability to reach a traditionally hard-to-reach population. Reasons for exclusion included 41 women who were over 26 weeks’ gestation, 149 who declined participation citing reasons such as being too busy, feeling unable to participate, perceiving no need for support, planning to move out of the area, or loss of further contact. In addition, six women experienced miscarriage or pregnancy loss, eight already had support elsewhere, and sixty-four were excluded due to insufficient English proficiency.

Randomisation and allocation: Randomisation procedures were successfully implemented, with 18 women randomised to the intervention group and 19 women to the usual care group. Four women in the usual care group met the clinical criteria and received the intervention. However, for ITT analysis, all participants were analysed in their originally assigned groups, preserving the benefits of randomisation. Group allocation integrity was otherwise maintained throughout the trial. The crossover of some control participants to the intervention highlights the practical feasibility of delivering flexible, responsive care and the importance of accommodating participants’ needs in real-world settings.

A detailed recruitment flowchart outlining eligibility assessment, randomisation, follow-up, and analysis is presented as Figure 1.

Intervention delivery and fidelity: The enhanced midwifery care was delivered with good fidelity. Participants in the intervention arm received coordinated and continuous support from a dedicated RM trained in counselling, who collaborated with a multidisciplinary team to address psychosocial and clinical needs. All participants in the intervention group adhered to their planned contacts with the RM, tailored to their individual needs. Despite the challenges posed by the COVID-19 pandemic, remote adaptations such as telehealth consultations and virtual check-ins (contacts ranging from 1 to 18 times) ensured consistent delivery of the intervention and maintained continuity of care. There were no reported injuries or adverse events during the intervention.

Case example of intervention: To demonstrate how the enhanced midwifery care model was applied in practice, one participant received a series of personalised contacts throughout pregnancy and postpartum. After allocation to the intervention group, Contact 1 oriented the participant to the service. Contact 2 provided evidence-based advice about COVID-19 vaccination, while Contacts 3–4 offered reassurance regarding common pregnancy-related discomforts such as abdominal pain and fatigue. Subsequent contacts addressed emerging needs, including information on waterbirth options (Contact 5), reassurance about a perceived leg bruise and guidance on persistent headaches (Contacts 6–7), and reminders about the availability of ongoing support (Contact 8). Later contacts involved anticipatory guidance on foetal movements (Contact 9), followed by postnatal support such as reassurance about neonatal jaundice (Contacts 10–11). Continued postpartum check-ins included breastfeeding and milk supply advice when planning travel (Contact 12). This example highlights the responsive, relationship-based nature of the intervention, with the RM adapting the frequency and content of contacts to meet the participant’s evolving needs across the perinatal period.

Retention and follow-up: Retention was moderate, with 12 of 18 (67%) participants in the intervention group and 17 of 19 (89%) in the control group retained at the 6-month follow-up. Although retention appeared higher in the control group, this difference was not statistically significant (Fisher’s exact test, *p* = 0.12), likely reflecting the small sample size.

### 3.3. Findings of the Linear Mixed-Effects Models

The results of the ITT linear mixed-effects models for EDS, K10, and KPCS scores are presented in Table 2 and Figure 2. The mixed-effects models showed a significant effect of time but no significant effect of treatment and no interaction between treatment and time for EDS scores. Further, there were no significant effects of time or treatment or a significant interaction between time and treatment group for K10 and KPCS (Table 2 and Figure 2).

Pairwise comparisons of EDS scores across timepoints showed a significant within-group reduction in depressive symptoms from baseline (T0) to 8 weeks post-partum (T2) (Table 3).

## 4. Discussion

This study assessed the feasibility and efficacy of enhanced midwifery care in improving depressive symptoms, psychological distress, and parenting confidence among pregnant women from ethnically diverse communities of SWSLHD. The enhanced midwifery care was delivered with high fidelity, including adaptations for remote support during the COVID-19 pandemic to ensure continuity of care. Retention at six months was reasonable, with 67% of participants in the intervention group and 89% in the control group completing follow-up assessments. Importantly, participants in both groups showed meaningful reductions in depressive symptoms over time, with within-group analyses indicating a significant improvement from baseline to 8 weeks post-partum.

This pilot trial demonstrated that enhanced midwifery care can be feasibly implemented among a traditionally hard-to-reach group of women from CALD background, achieving successful recruitment. However, a number of potentially eligible women were unable to participate due to limited English proficiency or relocation, and many declined participation due to being too busy, feeling unable to participate, or perceiving no need for support. These findings highlight the practical barriers encountered when engaging this population and underscore the importance of tailored recruitment strategies with use of interpreters. Addressing such barriers is crucial for ensuring more inclusive participation and provides valuable guidance for future research with priority populations.

The inclusion of telehealth consultations and virtual check-ins allowed continuity of care during the COVID-19 pandemic and demonstrated the adaptability of the enhanced midwifery care model to different service delivery contexts. Importantly, engagement remained strong, suggesting that the use of trusted providers such as midwives fostered rapport and trust, which are essential for sustained participation. Although retention was moderate, this rate is consistent with challenges observed in clinical trials involving women from CALD backgrounds, where competing life demands, access, and engagement barriers can affect continued participation [38,39]. Nevertheless, the ability to recruit and retain women from CALD backgrounds highlights the potential for scaling similar models of care in multicultural communities.

Despite no statistically significant improvements associated with the intervention, it was notable that, over time, both groups showed improvements in their mental health. There are several mechanisms which may have influenced the findings of this pilot study. First, it is important to consider that the study was mainly carried out during COVID-19 associated lockdowns, and therefore the health system was struggling to support women during pregnancy and post-partum stages. The heightened uncertainty experienced during this time likely increased the variability in the study sample relative to what might be expected under typical circumstances. Some families were better able to cope with socio-political pressures, which in turn may have made differences between groups harder to disentangle. Similarly, it is also possible that those in the control group received adequate support from nurses at the participating hospitals that were experienced in screening for comorbid mental health concerns, as this has been an area of focus in prior studies in the same setting [40].

It is also possible that women who chose to take part in the study were more likely than average clients to engage with health services and had an increased capacity to seek out supports regardless of the intervention received. This possibility arises given the large proportion of English-speaking and Australian-born families in the study. While there is limited data in relation to families that did not take part in the study, the final sample appeared to be less diverse than the average population of the area, especially in terms of the proportion of families speaking a language other than English at home (16–28% in the study sample versus 55% in South West Sydney) [41]. Indeed, priority population groups are generally less likely to access health services and participate in research [19]. It is possible the acceptability of the intervention, or of participation in a research study, was seen as less appetitive to non-Australian-born English-speaking families, as discussed in the Methods Section.

It is also worth considering that the 6-month follow-up period, while a strength of the study design, may not have been enough to fully capture the mental health journeys of participants. Note that differences between the intervention group and the control, while remaining non-significant, appear to show increased divergence at 6 months relative to earlier timepoints. It is possible that a longer follow-up period may elucidate larger discrepancies between control and intervention groups. A larger magnitude of differences between groups may have been present had mental health concerns been more severe at the start of the study. Considering the study focused on women with sub-clinical levels of postnatal depression, it may have been overly ambitious for the group to experience gains that moved the group average below the average population EDS score. Indeed, both groups’ mean EDS score changed from the “watch-and-wait” range to normative EDS levels [13]. A more substantial change may have been expected if baseline EDS scores were higher. This is an important consideration for studies investigating sub-clinical populations, where lower initial symptom severity can result in smaller effect sizes. Consequently, larger sample sizes may be required to achieve adequate power, compared to studies involving participants with more severe presentations.

Although there were no significant differences between the intervention and the control groups, this study provides important implications for public health practice and policy around enhanced midwifery care models that could be adapted in diverse health system contexts. Embedding systematic screening and tailored supports with continuity of care and integrating midwifery care within the multidisciplinary framework that incorporates a sociocultural lens during the perinatal period may help reduce the overall burden of antenatal distress at community level. As demonstrated in other studies, such approaches can lead to improved maternal, child, and family health and wellbeing, as well as better economic outcomes [42,43]. Although conducted in SWSLHD, the findings are relevant to international contexts where disadvantaged and CALD populations face similar challenges in accessing perinatal mental health support. This study adds to the growing evidence on the impact of models of care including enhanced midwifery care, demonstrating the feasibility and the challenges of implementation in underserved communities [44]. These insights may inform health systems in other countries experiencing similar challenges in perinatal mental health and allow them to consider scalable adaptations of the enhanced midwifery care model.

This study has several limitations. The small sample size, exacerbated by COVID-19-related disruptions and funding restrictions, limited the power to detect significant effects. Additionally, the pandemic may have introduced variability in maternal mental health outcomes, as women faced heightened uncertainty with access to services. The absence of other key information such as level of education, household income, medical history such as parity, and infant outcome data further limited the interpretation of findings. Additionally, recruiting women from CALD backgrounds was challenging due to language barriers, reluctance to engage with the research midwife, and limited trust in university branding. Although recruitment materials were translated and the study’s health system affiliation emphasised, these strategies had little impact. Future studies may benefit from targeting specific communities, availability of interpreters, matching the RM’s cultural background, and recruiting via trusted community partners. Additionally, future directions include expanding on this pilot study by using a larger and adequately powered trial with long-term follow-up are needed to determine the effectiveness and cost-effectiveness of the enhanced midwifery care models. Further, future research should specifically examine how cultural differences, including family support structures and traditional perinatal practices, may shape responses to enhanced midwifery care.

Despite these limitations, the study also has a number of strengths, including the random allocation of participants and the successful completion of automated questionnaires by participants across timepoints. This study targeted a population that is usually very difficult to reach, demonstrating priority populations can be engaged in research studies (although noting they may require longer recruitment periods and targeted strategies to achieve a higher recruitment rate). Finally, the measures used in the study were standardised, evidence-based measures recommended by clinical practice guidelines. Further, perspectives of care received by mothers and service providers in relation to the midwifery continuity of care model has been published elsewhere [44].

## 5. Conclusions

Findings of this pilot study demonstrated no statistically significant differences between the intervention and control group in improving depression or psychological distress as indicated by EDS and K10 scores, although parenting confidence improved across groups. Limitations such as small sample size, subclinical baseline symptoms, COVID-19-related disruptions, and difficulties engaging participants from CALD backgrounds may have constrained the detection of intervention effects. Future studies are needed that offer tailored, culturally safe, and linguistically appropriate supports which proactively address systemic barriers to care and ensure equitable access for families whose primary language is not English. Additionally, larger sample sizes are required when working with participants presenting with subclinical symptoms, supported by coordinated and continuity of care models to ensure equitable and high-quality support.

## Figures and Tables

**Figure 1 ijerph-22-01835-f001:**
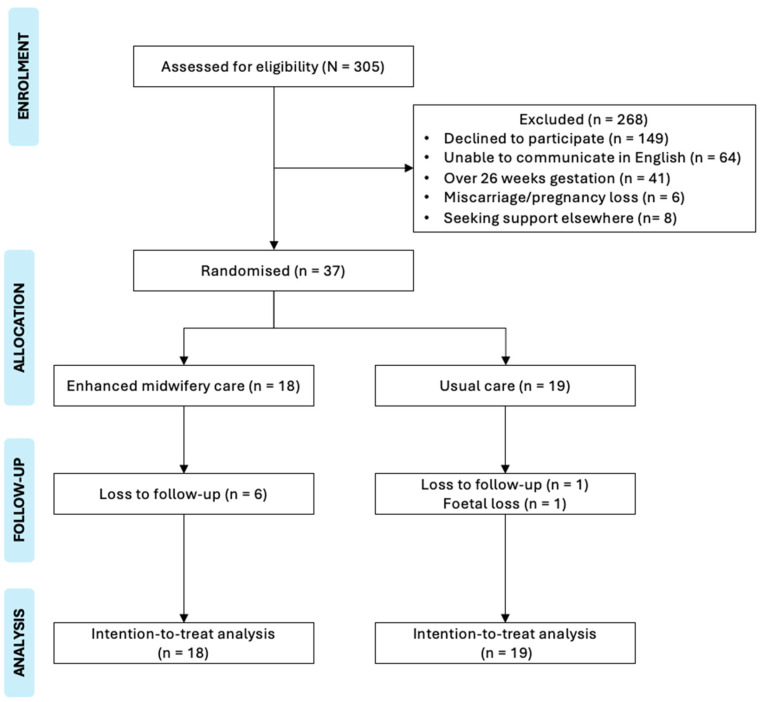
Recruitment flowchart.

**Figure 2 ijerph-22-01835-f002:**
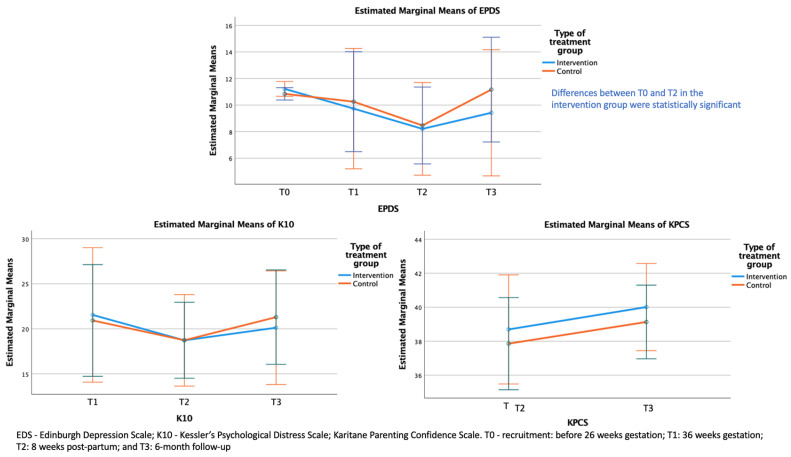
Estimated marginal means of EDS, K10, and KPCS across timepoints.

**Table 1 ijerph-22-01835-t001:** Baseline demographic and clinical characteristics by treatment groups.

Variables	Intervention Group (N = 18) n (%)	Control Group (N = 19) n (%)	*p*-Value
Mother’s age in years, mean (SD)	29.1 (7.1)	30.6 (6.0)	0.501
Number of weeks pregnant, mean (SD)	18.4 (4.5)	19.3 (3.5)	0.514
Country of birth			0.618
Australia	10 (55.6)	9 (47.4)	
Other	8 (44.4)	10 (52.6)	
Ethnicity			0.632
Caucasian	6 (33.3)	5 (26.3)	
Non-Caucasian	11 (61.1)	13 (68.4)	
Missing/Undisclosed	1 (5.6)	1 (5.3)	
Language spoken at home			0.800
English	13 (72.2)	16 (84.2)	
Other	5 (27.8)	3 (15.8)	
CALD status *			0.632
No	6 (33.3)	5 (26.3)	
Yes	11 (61.1)	13 (68.4)	
Missing/Undisclosed	1 (5.6)	1 (5.3)	
Indigenous status			0.604 ^a^
No	17 (94.4)	16 (84.2)	
Yes	1 (5.6)	3 (15.8)	
Smoking status			1.000 ^a^
No	17 (94.4)	18 (94.7)	
Yes	1 (5.6)	1 (5.3)	
EDS at baseline, mean (SD)	11.11 (0.83)	10.84 (0.77)	0.313
K10 at baseline, mean (SD)	19.50 (8.31)	19.63 (10.42)	0.973

^a^ Fisher’s exact test for variables with more than 20% of cells with expected counts below 5; * CALD status—non-Caucasian and/or language other than English; SD—standard deviation; EDS—Edinburgh Depression Scale; K10—Kessler’s Psychological Distress Scale.

**Table 2 ijerph-22-01835-t002:** Linear mixed-effects models.

Outcomes		β	SE	*p*-Value
EDS	Intercept	16.42	4.34	0.001
	Treatment (Reference = Control)	−0.02	1.63	0.992
	Time (Ref = T0)			
	T1	−1.50	1.39	0.283
	T2	−2.77	1.36	0.045
	T3	−1.14	1.39	0.414
	Time (Ref = T0) × Treatment (Ref = Control)			
	Time (T1) × Treatment (Intervention)	−1.15	2.08	0.581
	Time (T2) × Treatment (Intervention)	−1.34	2.10	0.524
	Time (T3) × Treatment (Intervention)	−1.80	2.08	0.389
K10 scale	Intercept	27.04	9.19	0.007
	Treatment (Ref = Control)	−0.58	3.20	0.856
	Time (Ref = T1)			
	T2	−1.20	1.93	0.535
	T3	1.06	1.97	0.593
	Time (Ref = T1) × Treatment (Ref = Control)			
	Time (T2) × Treatment (Intervention)	−1.05	3.02	0.729
	Time (T3) × Treatment (Intervention)	−1.99	2.99	0.509
KPCS	Intercept	34.30	4.65	<0.001
	Treatment (Ref = Control)	0.87	1.58	0.584
	Time (Ref = T2)			
	T3	1.44	1.03	0.176
	Time (Ref = T2) × Treatment (Ref = Control)			
	Time (T3) × Treatment (Intervention)	−0.31	1.60	0.847

Adjusted for covariates. T0—recruitment: before 26 weeks’ gestation; T1: 36 weeks’ gestation; T2: 8 weeks post-partum; T3: 6-month follow-up; β—coefficients; SE—standard error; Ref—reference group; EDS—Edinburgh Depression Scale; K10—Kessler’s Psychological Distress Scale; Karitane Parenting Confidence Scale.

**Table 3 ijerph-22-01835-t003:** Changes in EDS scores across timepoints.

Contrast	Estimate	SE	*p*-Value
T0–T1	2.08	1.04	0.199
T0–T2	3.44	1.05	0.008
T0–T3	2.04	1.04	0.212
T1–T2	1.36	1.11	0.613
T1–T3	−0.03	1.10	1.000
T2–T3	−1.40	1.11	0.594

T0—recruitment: before 26 weeks’ gestation; T1: 36 weeks’ gestation; T2: 8 weeks post-partum; T3: 6-month follow-up; SE—standard error.

## Data Availability

The data that support the findings of this study are not openly available due to reasons of sensitivity and are available from the corresponding author upon reasonable request.

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
