# Peer review of "Feasibility and Preliminary Efficacy of Enhanced Midwifery Care to Support Women Experiencing Subclinical Depression: A Pilot Randomised Controlled Trial"

_ijerph, 2025, doi:10.3390/ijerph22121835_

Round 1
Reviewer 1 Report
Comments and Suggestions for Authors
This pilot randomized controlled trial investigates the effect of enhanced midwifery care on reducing subclinical depression features in pregnant women from an ethnically diverse and socio-economically disadvantaged area in South Western Sydney. The study is well-structured and addresses a significant public health issue. Below is a detailed review based on the requested criteria, suggesting minor revisions.
- The authors mention that the RM received "light-touch training in mental health first aid strategies" in Methods. To improve replicability, it would be beneficial to briefly specify the duration or content of this "light-touch" training.
- In the discussion( 8 page) , the authors note that the final sample "appeared to be less diverse than the average population of the area," citing the large proportion of English-speaking and Australian-born families. To better contextualize the study's challenge in recruiting the target priority population, include the actual percentages of English speakers and Australian-born participants directly into the relevant paragraph in the Discussion.
.
Author Response
Point-by-point response to the reviewer’s comments attached as Word document.

Reviewer 2 Report
Comments and Suggestions for Authors
Dear authors
I appreciate the intent of the study and the focus on subclinical depression in a sample of expectant mothers. I have a few suggestions /observations for your consideration:
- The percentage of of women who declined to participate is substantial ( nearly 60%). It would very helpful if there is some information mentioned about the reasons for the same and the basic sociodemographic profile of the non consenting women. This may substantiate the arguments presented later on to understand the study results.
- Similarly, it is reported that 86 women did not meet the inclusion criteria. Would it be feasible to mention any dominant reasons in terms of the nature of criteria that were not met, resulting in exclusion?
- Sample size calculations are mentioned. But the authors do not mention the estimated sample size in an explicit manner.
- Each of the arms are described in detail separately. However, for a general reader, it may be good to summarize in a few lines as to how/in what specific ways the enhanced care group was different from the routine care group.
- Was there any feedback taken from the two groups about the nature of care received ?Could that have thrown some light on the perceptions of care and have implications for the findings? Can this limitation be added? Would it help if future studies focus on needs, expectations and actual experiences of care, qualitatively, in a study of this kind, rather than limiting to measures of depression and distress.
- Towards the end, the authors suggest specific interventions in further studies- but that seems contrary to the initial rationale for undertaking the study ( low intensity intervention delivered by a non specialist).
Author Response
Point-by-point response to the reviewer’s comments attached as a Word document.

Reviewer 3 Report
Comments and Suggestions for Authors
I attach a Word file with my detailed comments to the authors.

Author Response

(The authors gave the same response as above.)

Reviewer 4 Report
Comments and Suggestions for Authors
Please see the attachment

Author Response

(The authors gave the same response as above.)

Reviewer 5 Report
Comments and Suggestions for Authors
- Inclusion and, especially exclusion, criteria should be clearly listed in the same methods subsection.
- “Of the 305 participants assessed for eligibility, 37 mothers were eligible and consented to participate, with 18 women randomised to the intervention group and 19 women to the usual care group.” In the flow chart, it seems that 86 were not meeting the inclusion criteria, therefore it should be better clarified the definition of eligibility.
- Please, clarify if figure 2 panels reflect table 2. In this case, the statistically significant differences at the specific time points could be better highlighted.
- Table 3 should be explained better in both caption and text.
- I think in general the results could be expanded and described more in detail.
- The conclusion should be more schematic.
See comments above
Author Response

(The authors gave the same response as above.)

Round 2
Reviewer 2 Report
Comments and Suggestions for Authors
Dear Authors
Substantial revisions have been made in this version to improve the methodology presentation. I also see changes in objectives and inclusion of other details.
Suggestions for revision in the last paragraph as indicated below:
"Future studies are needed using specific tailored, culturally safe interventions supports that target women who do not have the capacity to seek out help such as non-English speaking women. Further, larger sample sizes are needed when targeting mid-range clinical acuity, with active support from several midwives to provide the best care possible."
I would suggest that the boldfaced wordings in the above paragraph are reconsidered in order to ensure equity-framing, non stigmatizing language that takes in account and avoiding individual deficit focused phrasing that seems to downplay systemic barriers which limit access to services. Moreover, the provision of linguistically appropriate support may represent an important factor to highlight.
Also, the last sentence is unclear ('mid range clinical acuity' and 'several midwives' ) and rephrasing may be considered.
Author Response

(The authors gave the same response as above.)

Reviewer 3 Report
Comments and Suggestions for Authors
I thank the authors for their careful consideration of my previous comments and their work in improving the manuscript at hand. I have a few remaining comments.
- Did the authors use statistics such as Fisher’s exact test or chi-square test to compare the primary outcomes, including, e.g., retention rate? If not, given that this is the main aim of the work, I suggest adding these.
- Could the authors please report more details how fidelity was measured?
- I feel that to understand the results well, it would be very helpful to understand the content of both interventions. The descriptions presented in the revision are greatly extended and improved, it now becomes much clearer what the intended differences between the interventions were. However, I feel that some more information on the dose and the content of the respective interventions that the women actually received – not that were planned – would be very relevant. This is especially true for the control group: Are any information available? For instance, how many times the women in the control group met, with whom they met (i.e., what profession), and how long their contacts were?
Author Response

(The authors gave the same response as above.)

Reviewer 5 Report
Comments and Suggestions for Authors
I have no additional major comments
Comments on the Quality of English LanguageSee comments above
Author Response
Comment 1: I have no additional major comments
Author response: We thank the reviewer for their time and assessment of our revised manuscript. We have carefully reviewed and edited the methods and results section to improve clarity.